# RETHINKING DECISION TRANSFORMER VIA HIERARCHICAL REINFORCEMENT LEARNING

## ABSTRACT

Decision Transformer (DT) is an innovative algorithm leveraging recent advances of the Transformer architecture in sequential decision making. However, a notable limitation of DT is its reliance on recalling trajectories from datasets, without the capability to seamlessly stitch them together. In this work, we introduce a general sequence modeling framework for studying sequential decision making through the lens of *Hierarchical Reinforcement Learning*. At the time of making decisions, a *high-level* policy first proposes an ideal *prompt* for the current state, a *low-level* policy subsequently generates an action conditioned on the given prompt. We show how DT emerges as a special case with specific choices of high-level and low-level policies and discuss why these choices might fail in practice. Inspired by these observations, we investigate how to jointly optimize the high-level and low-level policies to enable the stitching capability. This further leads to the development of new algorithms for offline reinforcement learning. Finally, our empirical studies clearly demonstrate the proposed algorithms significantly surpass DT on several control and navigation benchmarks. We hope that our contributions can inspire the integration of Transformer architectures within the field of RL.

## 1 INTRODUCTION

One of the most remarkable characteristics observed in large sequence models, especially Transformer models, is the *in-context learning* ability (Radford et al., 2019; Brown et al., 2020; Ramesh et al., 2021; Gao et al., 2020; Akyürek et al., 2022; Garg et al., 2022; Laskin et al., 2022; Lee et al., 2023). With the appropriate *prompt*, a pre-trained transformer can learn new tasks without explicit supervision and additional parameter updates. Decision Transformer (DT) is an innovative method that attempts to explore this idea for sequential decision making (Chen et al., 2021). Unlike traditional Reinforcement Learning (RL) algorithms, which learn a value function by bootstrapping or computing policy gradient, DT directly learns an autoregressive generative model from trajectory data $(R_0, s_0, a_0, \ldots, R_t, s_t, a_t)$ using a causal transformer (Vaswani et al., 2017; Radford et al., 2019). Here, $R_t$ is the *return-to-go*, which is the sum of future rewards along the trajectory starting from time step $t$. This approach allows leveraging existing transformer architectures widely employed in language and vision tasks that are easy to scale, and benefitting from a substantial body of research focused on stable training of transformer (Radford et al., 2019; Brown et al., 2020; Fedus et al., 2022; Chowdhery et al., 2022).

We argue that DT can be viewed as a model that *learns what action should be taken at a given state in order to make so many returns*. Following this, the return-to-go prompt is like a *switch*, guiding the model in making decisions at test time. If such a model can be learned effectively and generalized well even for out-of-distribution returns, it is reasonable to expect that DT can generate a better policy by prompting a higher return. Unfortunately, this seems to demand a level of generalization ability that is often too high in practical sequential decision-making problems. In fact, the key challenge facing DT is how to improve its robustness to the underlying data distribution, particularly when learning from trajectories collected by policies that are not close to optimal. Recent studies have indicated that for problems requiring the *stitching ability*, referring to the capability to integrate suboptimal trajectories from the data, DT cannot provide a significant advantage compared to behavior cloning (Fujimoto and Gu, 2021; Emmons et al., 2021; Kostrikov et al., 2022; Yamagata et al., 2023; Badrinath et al., 2023; Xiao et al., 2023a). This further confirms that a naive return-to-go prompt is not good enough for solving complex sequential decision-making problems.

Recent works on large language models demonstrate that carefully engineered prompts, either human-written or self-discovered by the model, significantly boost the performance of transformer models (Lester et al., 2021; Singhal et al., 2022; Zhang et al., 2022; Wei et al., 2022; Wang et al., 2022; Yao et al., 2023; Liu et al., 2023). In particular, it has been observed that the ability to perform complex

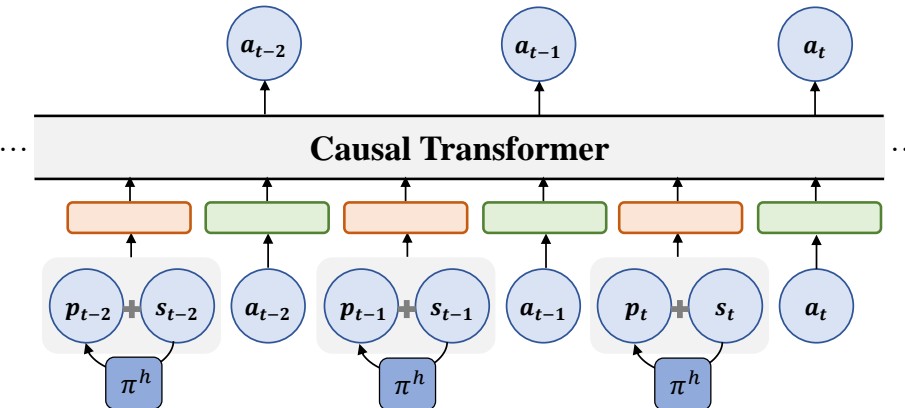

Figure 1: ADT architecture. The high-level policy generates a prompt, prompting the low-level policy to compute an action. In our implementation, prompts are concatenated with the states instead of being treated as single tokens. The concatenated vectors and actions are fed into linear embeddings with a positional episodic timestep encoding added, respectively. Tokens are fed into the causal transformer model which predicts actions autoregressively.

reasoning naturally emerges in sufficiently large language models when they are presented with a few chain of thought demonstrations as exemplars in the prompts (Wei et al., 2022; Wang et al., 2022; Yao et al., 2023). Driven by the significance of these works in language models, a question arises: *For RL, is it feasible to learn to generate prompts for transformer models to produce optimal control policies?* This paper attempts to address this problem. Our main contributions are:

- We present a generalized framework for studying decision-making through sequential modeling by connecting it with *Hierarchical Reinforcement Learning* (Nachum et al., 2018): a high-level policy first suggests a prompt for the current state, a low-level policy subsequently generates an action conditioned on the given prompt. We show DT can be recovered as a special case of this framework.

- We investigate when and why DT fails in terms of stitching sub-optimal trajectories. To overcome this drawback of transformer-based decision models, we investigate how to jointly optimize the high-level and low-level policies to enable the stitching capability. This further leads to the development of two new algorithms for offline reinforcement learning.

- We provide experiment results on several offline RL benchmarks, including locomotion control, navigation and robotics, to demonstrate the effectiveness of the proposed algorithms. Additionally, we conduct thorough ablation studies on the key components of our algorithms to gain deeper insights into their contributions. Through these ablation studies, we assess the impact of specific algorithmic designs on the overall performance.

## 2 PRELIMINARIES

### 2.1 OFFLINE REINFORCEMENT LEARNING

We consider Markov Decision Process (MDP) determined by $M = \{\mathcal{S}, \mathcal{A}, P, r, \gamma\}$ (Puterman, 2014), where $\mathcal{S}$ and $\mathcal{A}$ represent the state and action spaces. The discount factor is given by $\gamma \in [0, 1)$, $r : \mathcal{S} \times \mathcal{A} \to \mathbb{R}$ denotes the reward function, $P : \mathcal{S} \times \mathcal{A} \to \Delta(\mathcal{S})$ defines the transition dynamics[1]. Let $\tau = (s_0, a_0, r_0, \ldots, s_T, a_T, r_T)$ be a trajectory. Its *return* is defined as the discounted sum of the rewards along the trajectory: $R = \sum_{t=0}^{T} \gamma^t r_t$. Given a policy $\pi : \mathcal{S} \to \Delta(\mathcal{A})$, we use $\mathbb{E}^\pi$ to denote the expectation under the distribution induced by the interconnection of $\pi$ and the environment. The *value function* specifies the future discounted total reward obtained by following policy $\pi$,

$$V^\pi(s) = \mathbb{E}^\pi \left[ \sum_{t=0}^{\infty} \gamma^t r(s_t, a_t) \Big| s_0 = s \right], \tag{1}$$

There exists an *optimal policy* $\pi^*$ that maximizes values for all states $s \in \mathcal{S}$.

---

[1]We use $\Delta(\mathcal{X})$ to denote the set of probability distributions over $\mathcal{X}$ for a finite set $\mathcal{X}$.

In this work, we consider learning an optimal control policy from previously collected offline dataset, $\mathcal{D} = \{\tau_i\}_{i=0}^{n-1}$, consisting of $n$ trajectories. Each trajectory is generated by the following procedure: an initial state $s_0 \sim \mu_0$ is sampled from the initial state distribution $\mu_0$; for time step $t \geq 0$, $a_t \sim \pi_{\mathcal{D}}$, $s_{t+1} \sim P(\cdot|s_t, a_t), r_t = r(s_t, a_t)$, this process repeats until it reaches the maximum time step of the environment. Here $\pi_{\mathcal{D}}$ is an *unknown behavior policy*. In offline RL, the learning algorithm can only take samples from $\mathcal{D}$ without collecting new data from the environment (Levine et al., 2020).

## 2.2 DECISION TRANSFORMER

Decision Transformer (DT) is an extraordinary example that bridges sequence modeling with decision-making (Chen et al., 2021). It shows that a sequential decision-making model can be made through minimal modification to the transformer architecture (Vaswani et al., 2017; Radford et al., 2019). It considers the following trajectory representation that enables autoregressive training and generation:

$$\tau = \left( \widehat{R}_0, s_0, a_0, \widehat{R}_1, s_1, a_1, \ldots, \widehat{R}_T, s_T, a_T \right) . \tag{2}$$

Here $\widehat{R}_t = \sum_{i=t}^{T} r_i$ is the *returns-to-go* starting from time step $t$. We denote $\pi_{\mathrm{DT}}(a_t|s_t, \widehat{R}_t, \tau_t)$ the DT policy, where $\tau_t = (s_0, a_0, \widehat{R}_0, \ldots, s_{t-1}a_{t-1}, \widehat{R}_{t-1})^2$ is the sub-trajectory before time step $t$. As pointed and verified by Lee et al. (2023), $\tau_t$ can be viewed as as a *context* input of a policy, which fully takes advantages of the in-context learning ability of transformer model for better generalization (Akyürek et al., 2022; Garg et al., 2022; Laskin et al., 2022).

DT assigns a desired returns-to-go $R^0$, together with an initial state $s_0$ are used as the initialization input of the model. After executing the generated action, DT decrements the desired return by the achieved reward and continues this process until the episode reaches termination. Chen et al. (2021) argues that the conditional prediction model is able to perform policy optimization without using dynamic programming. However, recent works observe that DT often shows inferior performance compared to dynamic programming based offline RL algorithms when the offline dataset consists of sub-optimal trajectories (Fujimoto and Gu, 2021; Emmons et al., 2021; Kostrikov et al., 2022).

## 3 AUTOTUNED DECISION TRANSFORMER

In this section, we present *Autotuned Decision Transformer (ADT)*, a new transformer-based decision model that is able to stitch sub-optimal trajectories from the offline dataset. Our algorithm is derived based on a general hierarchical decision framework where DT naturally emerges as a special case. Within this framework, we discuss how ADT overcomes several limitations of DT by automatically tune the prompt for decision making.

## 3.1 KEY OBSERVATIONS

Our algorithm is derived by considering a general framework that bridges transformer-based decision models with hierarchical reinforcement learning (HRL) (Nachum et al., 2018). In particular, we use the following hierarchical representation of policy

$$\pi(a|s) = \int_{\mathcal{P}} \pi^h(p|s) \cdot \pi^l(a|s,p)dp , \tag{3}$$

where $\mathcal{P}$ is a set of prompts. To make a decision, the high-level policy $\pi^h$ first generates a prompt $p \in \mathcal{P}$, instructed by which the low-level policy $\pi^l$ returns an action conditioned on $p$. DT naturally fits into this hierarchical decision framework. Consider the following value prompting mechanism. At state $s \in \mathcal{S}$, the high-level policy generates a real-value prompt $R \in \mathbb{R}$, representing *"I want to obtain $R$ returns starting from $s$."*. Informed by this prompt, the low-level policy responses an action $a \in \mathcal{A}$, *"Ok, if you want to obtain returns $R$, you should take action $a$ now."*. This exactly what DT does. It applies a dummy high-level policy which initially picks a target return-to-go prompt and subsequently decrement it along the trajectory. The DT low-level policy, $\pi_{\mathrm{DT}}(\cdot|s, R, \tau)$, learns to predict which action to take at state $s$ in order to achieve returns $R$ given the context $\tau$.

To better understand the failure of DT given sub-optimal data, we re-examine the illustrative example shown in Figure 2 of Chen et al. (2021). The dataset comprises random walk trajectories and their associated per-state return-to-go. Suppose that the DT policy $\pi_{\mathrm{DT}}$ perfectly memorizes all trajectory

---

[2] We define $\tau_0$ the empty sequence for completeness.

information contained in the dataset. The return-to-go prompt in fact acts as a *switch* to guide the model to make decisions. Let $\mathcal{T}(s)$ be the set of trajectories starting from $s$ stored in the dataset, and $R(\tau)$ be the return of a trajectory $\tau$. Given $R' \in \{R(\tau), \tau \in \mathcal{T}(s)\}$, $\pi_{\mathrm{DT}}$ is able to output an action that leads towards $\tau$. Thus, given an *oracle return* $R^*(s) = \max_{\tau \in \mathcal{T}(s)} R(\tau)$, it is expected that DT is able to follow the optimal trajectory contained in the dataset following the switch.

There are several issues. *First*, the oracle return $R^*$ is not known. The initial return-to-go prompt of DT is picked by hand and might not be consistent with the one observed in the dataset. This requires the model to generalize well for unseen return-to-go and decisions. *Second*, even though $R^*$ is known for all states, memorizing trajectory information is still not enough for obtaining the stitching ability as $R^*$ only serves a lower bound on the maximum achievable return. To understand this, consider an example with two trajectories $a \to b \to c$, and $d \to b \to e$. Suppose that $e$ leads to a return of 10, while $c$ leads to a return of 0. In this case, using 10 as the return-to-go prompt at state $b$, DT should be able to switch to the desired trajectory. However, the information "leaning towards $c$ can achieve a return of 10" does not pass to $a$ during training, since the trajectory $a \to b \to e$ does not exist in the data. If the offline data contains another trajectory that starts from $a$ and leads to a mediocre return (e.g. 1), DT might switch to that trajectory at $a$ using 10 as the return-to-go prompt, missing a more promising path. Thus, making predictions conditioned on return-to-go alone is not enough for policy optimization. Some form of information backpropagation is still required.

## 3.2 Algorithms

ADT jointly optimizes the hierarchical policies to overcomes the limitations of DT discussed above. An illustration of ADT architecture is provided in Fig. 1. Similar to DT, ADT applies a transformer model for the low-level policy. Instead of (2), it considers the following trajectory representation,

$$\tau = (p_0, s_0, a_0, p_1, s_1, a_1, \ldots, p_T, s_T, a_T) . \qquad (4)$$

Here $p_i$ is the prompt generated by the high-level policy $p_i \sim \pi^h(\cdot|s_i)$, replacing the return-to-go prompt used by DT. That is, for each trajectory in the offline dataset, we relabel it by adding a prompt generated by the high-level policies for each transition. Armed with this general hierarchical decision framework, we propose two algorithms that apply different high-level prompting generation strategy while sharing a unified low-level policy optimization framework. We learn a high-level policy $\pi_\omega \approx \pi^h$ with parameters $\phi$, and a low-level policy $\pi_\theta \approx \pi^l$ with parameters $\theta$.

### 3.2.1 Value-prompted Autotuned Decision Transformer

Our first algorithm, *Value-promped Autotuned Decision Transformer (V-ADT)*, uses scalar values as prompts. But unlike DT, it applies a more principled design of value prompts instead of return-to-go. V-ADT aims to answer two questions: what is the maximum achievable value starting from a state $s$, and what action should be taken to achieve such a value? To answer these, we view the offline dataset $\mathcal{D}$ as an *empirical MDP*, $M_\mathcal{D} = \{\mathcal{S}_\mathcal{D}, \mathcal{A}, P_\mathcal{D}, r, \gamma\}$, where $\mathcal{S}_\mathcal{D} \subseteq \mathcal{S}$ is the set of observed states in the data, $P_\mathcal{D}$ is the transition, which is an empirical estimation of the original transition $P$ (Fujimoto et al., 2019). The optimal value of this empirical MDP is

$$V_\mathcal{D}^*(s) = \max_{a:\pi_\mathcal{D}(a|s)>0} r(s,a) + \gamma \mathbb{E}_{s' \sim P_\mathcal{D}(\cdot|s,a)} [V_\mathcal{D}^*(s')] . \qquad (5)$$

Let $Q_\mathcal{D}^*(s,a)$ be the corresponding state-action value. $V_\mathcal{D}^*$ is known as the *in-sample optimal value* in offline RL (Fujimoto et al., 2018; Kostrikov et al., 2022; Xiao et al., 2023b). Computing this value requires to perform dynamic programming without querying out-of-distribution actions. We apply Implicit Q-learning (IQL) to learn $V_\phi \approx V_\mathcal{D}^*$ and $Q_\psi \approx Q_\mathcal{D}^*$ with parameters $\phi, \psi$ (Kostrikov et al., 2022). Details of IQL are presented in the Appendix. We now describe how V-ADT jointly optimizes high and low level policies to facilitate stitching.

**High-Level policy** V-ADT considers $\mathcal{P} = \mathbb{R}$ and adopts a deterministic policy $\pi_\omega : \mathcal{S} \to \mathbb{R}$, which predicts the in-sample optimal value $\pi_\omega \approx V_\mathcal{D}^*$. Since we already have an approximated in-sample optimal value $V_\phi$, we let $\pi_\omega = V_\phi$. This high-level policy offers two key advantages. *First*, this approach efficiently facilitates information backpropagation towards earlier states on a trajectory, addressing a major limitation of DT. This is achieved by using $V_\mathcal{D}^*$ as the value prompt, ensuring that we have precise knowledge of the maximum achievable return for any state. Making predictions conditioned on $R^*(s)$ is not enough for policy optimization, since $R^*(s) = \max_{\tau \in \mathcal{T}(s)} R(\tau)$ only gives a lower bound on $V_\mathcal{D}^*(s)$ and thus would be a weaker guidance (see Section 3.1 for detailed discussions). *Second*, the definition of $V_\mathcal{D}^*$ exclusively focuses on the optimal value derived from observed data and thus avoids out-of-distribution returns. This prevents the low-level policy from making decisions conditioned on prompts that require extrapolation.

**Low-Level policy** Directly training the model to predict the trajectory, as done in DT, is not suitable for our approach. This is because the action $a_t$ observed in the data may not necessarily correspond to the action at state $s_t$ that leads to the return $V_{\mathcal{D}}^*(s_t)$. However, the probability of selecting $a_t$ should be proportional to the value of this action. Thus, we use *advantage-weighted regression* to learn the low-level policy (Peng et al., 2019; Kostrikov et al., 2022; Xiao et al., 2023b): given trajectory data (4) the objective is defined as

$$\mathcal{L}(\theta) = -\sum_{t=0}^{T} \exp\left(\frac{Q_\psi(s_t, a_t) - V_\phi(s_t)}{\alpha}\right) \log \pi_\theta(a_t|s_t, \pi_\omega(s_t)),  \tag{6}$$

where $\alpha > 0$ is a hyper-parameter. The low-level policy takes the output of high-level policy as input. This guarantees no discrepancy between train and test value prompt used by the policies. We note that the only difference of this compared to the standard maximum log-likelihood objective for sequence modeling is to apply a weighting for each transition. One can easily implement this with trajectory data for a transformer. In practice we also observe that the tokenization scheme when processing the trajectory data affects the performance of ADT. Instead of treating the prompt $p_t$ as a single token as in DT, we find it is beneficial to concatenate $p_t$ and $s_t$ together and tokenize the concatenated vector. We provide an ablation study on this in Section 5.4.3. This completes the description of V-ADT.

### 3.2.2 GOAL-PROMPTED AUTOTUNED DECISION TRANSFORMER

In HRL, the high-level policy often considers a latent action space. Typical choices of latent actions includes *sub-goal* (Nachum et al., 2018; Park et al., 2023), *skills* (Ajay et al., 2020; Jiang et al., 2022), and *options* (Sutton et al., 1999; Bacon et al., 2017; Klissarov and Machado, 2023). We consider goal-reaching problem as an example and use sub-goals as latent actions, which leads to our second algorithm, *Goal-promped Autotuned Decision Transformer (G-ADT)*. Let $\mathcal{G}$ be the goal space[3]. The goal-conditioned reward function $r(s, a, g)$ provides the reward of taking action $a$ at state $s$ for reaching the goal $g \in \mathcal{G}$. Let $V(s, g)$ be the universal value function defined by the goal-conditioned rewards (Nachum et al., 2018; Schaul et al., 2015). Similarly, we define $V_{\mathcal{D}}^*(s, g)$ and $Q_{\mathcal{D}}^*(s, a, g)$ the in-sample optimal universal value function. We also train $V_\phi \approx V_{\mathcal{D}}^*$ and $Q_\psi \approx Q_{\mathcal{D}}^*$ to approximate the universal value functions. We now describe how G-ADT jointly optimizes the policies.

**High-Level policy** G-ADT considers $\mathcal{P} = \mathcal{G}$ and uses a high-level policy $\pi_\omega : \mathcal{S} \to \mathcal{G}$. To find a shorter path, the high-level policy $\pi_\omega$ generates a sequence of sub-goals $g_t = \pi_\omega(s_t)$ that guides the learner step-by-step towards the final goal. We use a sub-goal that lies in $k$-steps further from the current state, where $k$ is a hyper-parameter of the algorithm tuned for each domain (Badrinath et al., 2023; Park et al., 2023). In particular, given trajectory data (4), the high-level policy learns the optimal *k-steps jump* using the recently proposed Hierarchical Implicit Q-learning (HIQL) algorithms (Park et al., 2023):

$$\mathcal{L}(\phi) = -\sum_{t=0}^{T} \exp\left(\frac{\sum_{t'=t}^{k-1} \gamma^{t'-t} r(s_{t'}, a_{t'}, g) + \gamma^k V_\phi(s_{t+k}, g) - V_\phi(s_t, g)}{\alpha}\right) \log \pi_\omega(s_{t+k}|s_t, g).$$

**Low-Level policy** The low-level policy in G-ADT learns to reach the sub-goal generated by the high-level policy. G-ADT shares the same low-level policy objective as V-ADT. Given trajectory data (4), it considers the following

$$\mathcal{L}(\theta) = -\sum_{t=0}^{T} \exp\left(\frac{Q_\psi(s_t, a_t, \pi_\omega(s_t)) - V_\phi(s_t, \pi_\omega(s_t))}{\alpha}\right) \log \pi_\theta(a_t|s_t, \pi_\omega(s_t)),$$

Note that this is exactly the same as (6) except that the advantages are computed by universal value functions. G-ADT also applies the same tokenization method as V-ADT by first concatenating $\pi_\omega(s_t)$ and $s_t$ together. This concludes the description of the G-ADT algorithm.

## 4 DISCUSSIONS

**Types of Prompts** Xu et al. (2022) introduces Prompt-DT, which leverages the sequential modeling ability of the Transformer architecture, using expert trajectory prompts as task-specific guides to adapt

---

[3]The goal space and state space could be the same (Nachum et al., 2018; Park et al., 2023)

to unseen tasks without extra finetuning. Reed et al. (2022) have delved into the potential scalability of transformer-based decision models through prompting. They show that a causal transformer, trained on multi-task offline datasets, showcases remarkable adaptability to new tasks through fine-tuning. The adaptability is achieved by providing a sequence prompt as the input of the transformer model, typically represented as a trajectory of expert demonstrations. Unlike such expert trajectory prompts, our prompt can be seen as a latent action generated by the high-level policy, serving as guidance for the low-level policy to inform its decision-making process.

**Comparison of other DT Enhancements** Several recent works have been proposed to overcome the limitations of DT. Correia and Alexandre (2022) employs a dual transformer architecture to design Hierarchical DT (HDT), where a high-level mechanism selects sub-goal states from demonstration data to guide a low-level controller in task completion to improve DT. Yamagata et al. (2023) relabelled trajectory data by replacing return-to-go with values learned by offline RL algorithms. Badrinath et al. (2023) proposed to use sub-goal as prompt, guiding the DT policy to find shorter path in navigation problems. Wu et al. (2023) learned maximum achievable returns, $R^*(s) = \max_{\tau \in \mathcal{T}(s)} R(\tau)$, to boost the stitching ability of DT at decision time. Liu and Abbeel (2023) structured trajectory data by relabelling the target return for each trajectory as the maximum total reward within a sequence of trajectories. Their findings showed that this approach enabled a transformer-based decision model to improve itself during both training and testing time. Compared to these previous efforts, ADT introduces a principled framework of hierarchical policy optimization. Our practical studies show that the joint optimization of high and low level policies is the key to boost the performance of transformer-based decision models.

## 5 EXPERIMENT

We investigate three primary questions in our experiments. *First*, how well does ADT perform on offline RL tasks compared to prior DT-based methods? *Second*, is it essential to auto-tune prompts for transformer-based decision model? *Three*, what is the influence of various implementation details within an ADT on its overall performance? We refer readers to **??** for additional details and supplementary experiments.

### 5.1 EXPERIMENTAL SETTINGS

**Benchmark Problems** We leverage datasets across several domains including Gym-MuJoCo, AntMaze, and FrankaKitchen from the offline RL benchmark D4RL (Fu et al., 2020). For MuJoCo, we incorporate nine version 2 (v2) datasets. These datasets are generated using three distinct behavior policies: '-medium', '-medium-play', and '-medium-expert', and span across three specific tasks: 'halfcheetah', 'hopper', and 'walker2d'. The primary objective in long-horizon navigation task AntMaze is to guide an 8-DoF Ant robot from its starting position to a predefined target location. For this, we employ six version 2 (v2) datasets which include '-umaze', '-umaze-diverse', '-medium-play', 'medium-diverse', '-large-play', and '-large-diverse'. The Kitchen domain focuses on accomplishing four distinct subtasks using a 9-DoF Franka robot. We utilize three version 0 (v0) datasets that capture a range of behaviors: '-complete', '-partial', and '-mixed' for this domain.

**Baselines** We compare the performance of ADT with several representative baselines including (1) *offline RL methods*: TD3+BC (Fujimoto and Gu, 2021), CQL (Kumar et al., 2020) and IQL (Kostrikov et al., 2022); (2) *valued-conditioned methods*: Decision Transformer (DT) (Chen et al., 2021) and Q-Learning Decision Transformer (QLDT) (Yamagata et al., 2023); (3) *goal-conditioned methods*: HIQL (Park et al., 2023), RvS (Emmons et al., 2021) and Waypoint Transformer (WT) (Badrinath et al., 2023). All the baseline results except for QLDT are obtained from (Badrinath et al., 2023) and (Park et al., 2023) or by running the codes of CORL repository (Tarasov et al., 2022). For HIQL, we present HIQL's performance with the goal representation in Kitchen and that without goal representation in AntMaze, as per our implementation in ADT, to ensure fair comparison. QLDT and the transformer-based actor of ADT are implemented based on the DT codes in CORL, with similar architecture. Details are given in Appendix. The critics and the policies to generate prompts used in ADT are re-implemented in PyTorch following the official codes of IQL and HIQL. In all conducted experiments, five distinct random seeds are employed. Results are depicted with 95% confidence intervals, represented by shaded areas in figures and expressed as standard deviations in tables. The reported results of ADT in tables are obtained by evaluating the final models.

Table 1: Average normalized scores of V-ADT, value-conditioned (DT and QLDT), and value-based RL methods. The methods on the right of the vertical line are DT-based methods. The top scores among all DT-based methods are highlighted in **bold**.

| Environment | TD3+BC | CQL | IQL | DT | QLDT | V-ADT |
|---|---|---|---|---|---|---|
| halfcheetah-medium-v2 | $48.3 \pm 0.3$ | $44.0 \pm 5.4$ | $47.4 \pm 0.2$ | $42.4 \pm 0.2$ | $42.3 \pm 0.4$ | $\mathbf{48.7 \pm 0.2}$ |
| hopper-medium-v2 | $59.3 \pm 4.2$ | $58.5 \pm 2.1$ | $66.2 \pm 5.7$ | $63.5 \pm 5.2$ | $\mathbf{66.5 \pm 6.3}$ | $60.6 \pm 2.8$ |
| walker2d-medium-v2 | $83.7 \pm 2.1$ | $72.5 \pm 0.8$ | $78.3 \pm 8.7$ | $69.2 \pm 4.9$ | $67.1 \pm 3.2$ | $\mathbf{80.9 \pm 3.5}$ |
| halfcheetah-medium-replay-v2 | $44.6 \pm 0.5$ | $45.5 \pm 0.5$ | $44.2 \pm 1.2$ | $35.4 \pm 1.6$ | $35.6 \pm 0.5$ | $\mathbf{42.8 \pm 0.2}$ |
| hopper-medium-replay-v2 | $60.9 \pm 18.8$ | $95.0 \pm 6.4$ | $94.7 \pm 8.6$ | $43.3 \pm 23.9$ | $52.1 \pm 20.3$ | $\mathbf{83.5 \pm 9.5}$ |
| walker2d-medium-replay-v2 | $81.8 \pm 5.5$ | $77.2 \pm 5.5$ | $73.8 \pm 7.1$ | $58.9 \pm 7.1$ | $58.2 \pm 5.1$ | $\mathbf{86.3 \pm 1.4}$ |
| halfcheetah-medium-expert-v2 | $90.7 \pm 4.3$ | $91.6 \pm 2.8$ | $86.7 \pm 5.3$ | $84.9 \pm 1.6$ | $79.0 \pm 7.2$ | $\mathbf{91.7 \pm 1.5}$ |
| hopper-medium-expert-v2 | $98.0 \pm 9.4$ | $105.4 \pm 6.8$ | $91.5 \pm 14.3$ | $100.6 \pm 8.3$ | $94.2 \pm 8.2$ | $\mathbf{101.6 \pm 5.4}$ |
| walker2d-medium-expert-v2 | $110.1 \pm 0.5$ | $108.8 \pm 0.7$ | $109.6 \pm 1.0$ | $89.6 \pm 38.4$ | $101.7 \pm 3.4$ | $\mathbf{112.1 \pm 0.4}$ |
| gym-avg | $75.3 \pm 4.9$ | $77.6 \pm 3.4$ | $76.9 \pm 5.8$ | $65.3 \pm 10.1$ | $66.3 \pm 6.1$ | $\mathbf{78.7 \pm 2.8}$ |
| antmaze-umaze-v2 | 78.6 | 74.0 | $87.5 \pm 2.6$ | $53.6 \pm 7.3$ | $67.2 \pm 2.3$ | $\mathbf{88.2 \pm 2.5}$ |
| antmaze-umaze-diverse-v2 | 71.4 | 84.0 | $62.2 \pm 13.8$ | $42.2 \pm 5.4$ | $\mathbf{62.1 \pm 1.6}$ | $58.6 \pm 4.3$ |
| antmaze-medium-play-v2 | 10.6 | 61.2 | $71.2 \pm 7.3$ | $0.0 \pm 0.0$ | $0.0 \pm 0.0$ | $\mathbf{62.2 \pm 2.5}$ |
| antmaze-medium-diverse-v2 | 3.0 | 53.7 | $70.0 \pm 10.9$ | $0.0 \pm 0.0$ | $0.0 \pm 0.0$ | $\mathbf{52.6 \pm 1.4}$ |
| antmaze-large-play-v2 | 0.2 | 15.8 | $39.6 \pm 5.8$ | $0.0 \pm 0.0$ | $0.0 \pm 0.0$ | $\mathbf{16.6 \pm 2.9}$ |
| antmaze-large-diverse-v2 | 0.0 | 14.9 | $47.5 \pm 9.5$ | $0.0 \pm 0.0$ | $0.0 \pm 0.0$ | $\mathbf{36.4 \pm 3.6}$ |
| antmaze-avg | 27.3 | 50.6 | $63.0 \pm 8.3$ | $16.0 \pm 2.1$ | $21.6 \pm 0.7$ | $\mathbf{52.4 \pm 2.9}$ |
| kitchen-complete-v0 | $25.0 \pm 8.8$ | 43.8 | 62.5 | $46.5 \pm 3.0$ | $38.8 \pm 15.8$ | $\mathbf{55.1 \pm 1.4}$ |
| kitchen-partial-v0 | $38.3 \pm 3.1$ | 49.8 | 46.3 | $31.4 \pm 19.5$ | $36.9 \pm 10.7$ | $\mathbf{46.0 \pm 1.6}$ |
| kitchen-mixed-v0 | $45.1 \pm 9.5$ | 51.0 | 51.0 | $25.8 \pm 5.0$ | $17.7 \pm 9.5$ | $\mathbf{46.8 \pm 6.3}$ |
| kitchen-avg | $36.1 \pm 7.1$ | 48.2 | 53.3 | $34.6 \pm 9.2$ | $30.5 \pm 12.0$ | $\mathbf{49.3 \pm 3.1}$ |
| average | 52.7 | 63.7 | 68.3 | $43.8 \pm 7.3$ | $45.4 \pm 5.3$ | $\mathbf{65.0 \pm 2.9}$ |

Table 2: Average normalized scores of G-ADT and goal-conditioned methods. The methods on the right of the vertical line are DT-based methods. The top scores among all DT-based methods are highlighted in **bold**.

| Environment | RvS-R/G | HIQL | WT | G-ADT |
|---|---|---|---|---|
| antmaze-umaze-v2 | $65.4 \pm 4.9$ | $83.9 \pm 5.3$ | $64.9 \pm 6.1$ | $\mathbf{83.8 \pm 2.3}$ |
| antmaze-umaze-diverse-v2 | $60.9 \pm 2.5$ | $87.6 \pm 4.8$ | $71.5 \pm 7.6$ | $\mathbf{83.0 \pm 3.1}$ |
| antmaze-medium-play-v2 | $58.1 \pm 12.7$ | $89.9 \pm 3.5$ | $62.8 \pm 5.8$ | $\mathbf{82.0 \pm 1.7}$ |
| antmaze-medium-diverse-v2 | $67.3 \pm 8.0$ | $87.0 \pm 8.4$ | $66.7 \pm 3.9$ | $\mathbf{83.4 \pm 1.9}$ |
| antmaze-large-play-v2 | $32.4 \pm 10.5$ | $87.3 \pm 3.7$ | $\mathbf{72.5 \pm 2.8}$ | $71.0 \pm 1.3$ |
| antmaze-large-diverse-v2 | $36.9 \pm 4.8$ | $81.2 \pm 6.6$ | $\mathbf{72.0 \pm 3.4}$ | $65.4 \pm 4.9$ |
| antmaze-avg | $53.5 \pm 7.2$ | $86.2 \pm 5.4$ | $68.4 \pm 4.9$ | $\mathbf{78.1 \pm 2.5}$ |
| kitchen-complete-v0 | $50.2 \pm 3.6$ | $43.8 \pm 19.5$ | $49.2 \pm 4.6$ | $\mathbf{51.4 \pm 1.7}$ |
| kitchen-partial-v0 | $51.4 \pm 2.6$ | $65.0 \pm 9.2$ | $63.8 \pm 3.5$ | $\mathbf{64.2 \pm 5.1}$ |
| kitchen-mixed-v0 | $60.3 \pm 9.4$ | $67.7 \pm 6.8$ | $\mathbf{70.9 \pm 2.1}$ | $69.2 \pm 3.3$ |
| kitchen-avg | $54.0 \pm 5.2$ | $58.8 \pm 11.8$ | $61.3 \pm 3.4$ | $\mathbf{61.6 \pm 3.4}$ |
| average | $53.7 \pm 6.5$ | $77.1 \pm 7.5$ | $66.0 \pm 4.4$ | $\mathbf{72.6 \pm 2.8}$ |

## 5.2 MAIN RESULTS

In Table 1 and 2, we present the performance of two variations of ADT evaluated on offline datasets. Specifically, in Table 1, V-ADT excels, recording the most superior performance among all DT-based strategies. Notably, V-ADT surpasses two widely-used value-based offline RL techniques in overall. In the comparison with the value-conditioned transformer approaches, namely DT and QLDT, the V-ADT method exhibits significant superiority, particularly when evaluated on the sophisticated AntMaze and Kitchen datasets. Meanwhile, in Table 2, G-ADT notably exceeds the performance of other goal-conditioned approaches. To the best of our knowledge, this represents a benchmark that surpasses any prior DT-based methods. Given that V-ADT and G-ADT is trained following the IQL and HIQL paradigm, respectively, the achieved performance nearing or inferior to that of IQL and HIQL is anticipated.

An integration of findings from Table 1 and 2 further suggests that goal prompts might possess a comparative advantage over value prompts. One plausible explanation is that goal prompts assist in simplifying policy training by decomposing intricate tasks into manageable subtasks. Conversely, while value prompt might attempt to stitch sub-optimal trajectories, it primary focus on the overarching task. Anyhow, as both V-ADT and G-ADT outperforms DT, we can conclude that with appropriately crafted prompts and corresponding training regime as does in ADT, the capabilities of DT can be more effectively exploited to achieve better performance.

## 5.3 IMPACT OF MANUAL PROMPT TUNING ON DT PERFORMANCE

The prompt, i.e., target return, used by DT is a tunable hyper-parameter. Figure 2 delineates the results of DT using different target returns on four different walker2d datasets. The x-axis of each subfigure

represents the normalized target return input into DT, while the y-axis portrays the corresponding evaluation performance. Empirical results indicate that manual modifications to the target return could not improve the performance of DT, with its performance persistently lagging behind V-ADT. We also note that there is no single prompt that performs universally well on all problems. It is imperative to highlight that the utility of prompt in DT appears constrained, particularly when working with datasets sourced from unimodal behavior policy.

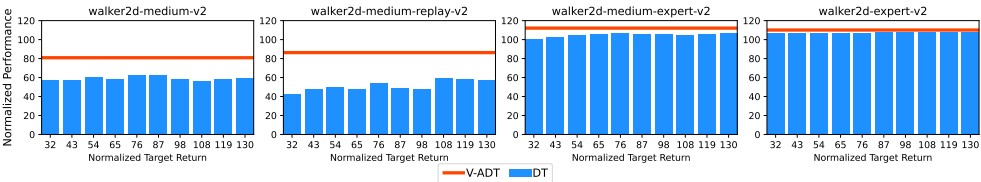

Figure 2: Average normalized results of DT using different prompt. Incorporating manual prompt engineering could not help DT outperform V-ADT.

## 5.4 ABLATION STUDIES ON ALGORITHM DESIGN CHOICES

### 5.4.1 IS PROMPT REALLY USEFUL?

In our earlier discussions, we posit that the benefits of using prompts in DT might be limited. Here we further investigate the efficacy of the prompt used by ADT. As the goal prompt in G-ADT is a necessary input, we only investigate on V-ADT. Referencing Fig.3, we compare the performance of V-ADT both with and without value prompt. The results indicate that value prompt may not be beneficial in certain environments, yet they show marked utility in others.

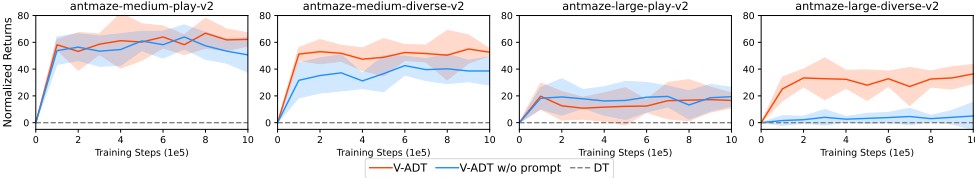

Figure 3: Learning curves of V-ADT with and without using prompt as part of input. The value prompt may not be beneficial in certain environments, yet they show marked utility in others.

### 5.4.2 CAN WE LEARN THE LOW-LEVEL POLICY AS IN DT?

We delve into the role of the loss Eq. (6) used by ADT. Figure 4 elucidates a comparison between V-ADT and G-ADT, both with and without the incorporation of the this loss. In the absence of Eq. (6), the method reverts back to using the prompt-conditioned behavior cloning loss, akin to the conventional DT approach. Our results show the substantial improvement in performance of both V-ADT and G-ADT when the loss Eq. (6) is leveraged. Intuitively, in the context where DT can entirely memorize the trajectories from the dataset and when the value prompt is fully accurate, using behavior cloning loss is expected to give DT the capability to stitch these trajectories seamlessly. However, ensuring absolute accuracy of the value prompt remains a challenge. As a result, even when DT recalls all trajectories through behavioral cloning, it cannot generalize to stitch trajectories under erroneous prompts. Moreover, even the obtained prompt is exactly accurate, the corresponding well-performed trajectories are probably absent in the dataset, hindering DT from adopting correct actions, as discussed before. Thus incorporating RL loss during the training of DT is necessary to enhance DT's trajectory stitching ability. Empirical results of V-ADT further corroborate this observation, meanwhile proving the previous finding that benefits of using prompts in DT might be limited. As for G-ADT, its performance is also notably compromised without the RL loss, yet it retains a performance edge over DT, attributed to the decomposition of the original task.

### 5.4.3 EFFICACY OF TOKENIZATION STRATEGIES

In ADT, we diverge from the methodology presented in (Chen et al., 2021) where individual tokens are produced for each input component: return-to-go prompt, state, and action. Instead, we opt for

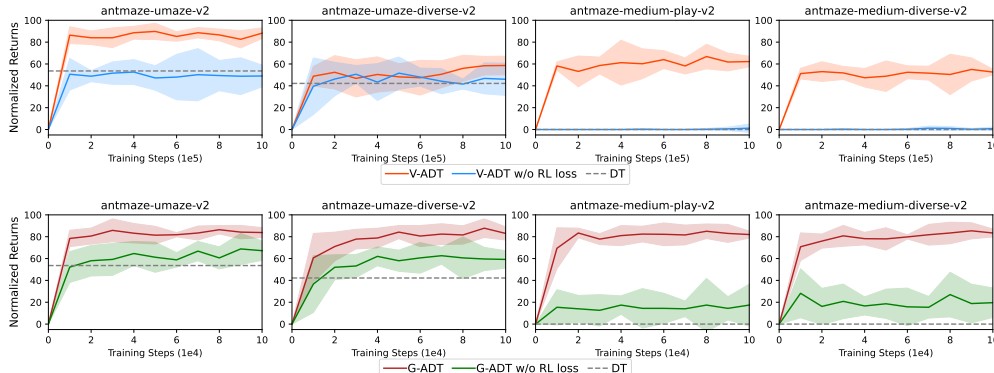

Figure 4: Learning curves of V-ADT and G-ADT with and without using RL loss. When RL loss is not used, the behavior cloning loss is used as does in DT. Results demonstrate that RL loss is essential in enpowering DT with stitching ability to achieve superior performance.

a concatenated representation of prompts and states. We present a comparative analysis between these two tokenization strategies in Fig. 5. It is evident that our token design contributes to superior performance in ADT, especially in G-ADT. We postulate that this is attributed to the design of high-policy, which ensures a high degree of correlation between states and the corresponding ideal prompts. Thus we assert that the states and the corresponding prompts should be treated with equal significance when computing attention within the transformer's internal architecture.

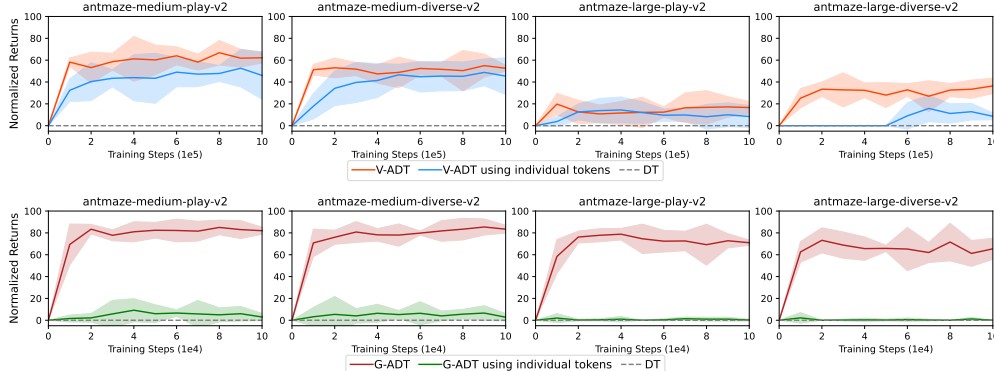

Figure 5: Learning curves of ADT with different tokenization strategies. Our design contributes to superior performance by equally treating the states and related prompts in computing attention.

## 6 CONCLUSIONS AND FUTURE WORKS

In this paper, we rethink transformer-based decision models through a hierarchical decision-making framework, where a high-level policy suggests a prompt, following which a low-level policy acts based on this suggestion. This allows us to investigate how to jointly optimize the high and low level policies to boost the model's performance when learning from sub-optimal data, which leads to the development of two new transformer-based decision models for offline RL, Value-prompted Auto-tuned DT and Goal-prompted Autotuned DT. Our results, complemented by in-depth ablation studies, underscore the effectiveness and innovation of our proposed methods. These findings not only resolve existing challenges in DT, but also pave the way for further exploration of Transformer architectures in reinforcement learning. There are several potential future research directions, including: First, investigating a wider variety of prompts to guide the optimal control of transformer-based policies to further enhance the generality of our framework; Second, applying the proposed framework in learning multi-modal and multi-task policies holds potential significance in designing foundation decision-making models.

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

# A  IMPLEMENTATION DETAILS

## A.1  ENVIRONMENTS

**MuJoCo**  For the MuJoCo framework, we incorporate nine version 2 (v2) datasets. These datasets are generated using three distinct behavior policies: '-medium', '-medium-play', and '-medium-expert', and span across three specific tasks: 'halfcheetah', 'hopper', and 'walker2d'.

**AntMaze**  The AntMaze represents a set of intricate, long-horizon navigation challenges. This domain uses the same umaze, medium, and large mazes from the Maze2D domain, but replaces the agent with an 8-DoF Ant robot from the OpenAI Gym MuJoCo benchmark. For the 'umaze' dataset, trajectories are generated with the Ant robot starting and aiming for fixed locations. To introduce complexity, the "diverse" dataset is generated by selecting random goal locations within the maze, necessitating the Ant to navigate from various initial positions. Meanwhile, the "play" dataset is curated by setting specific, hand-selected initial and target positions, adding a layer of specificity to the task. We employ six version 2 (v2) datasets which include '-umaze', '-umaze-diverse', '-medium-play', '-medium-diverse', '-large-play', and '-large-diverse' in our experiments.

**Franka Kitchen**  In the Franka Kitchen environment, the primary objective is to manipulate a set of distinct objects to achieve a predefined state configuration using a 9-DoF Franka robot. The environment offers multiple interactive entities, such as adjusting the kettle's position, actuating the light switch, and operating the microwave and cabinet doors, inclusive of a sliding mechanism for one of the doors. For the three principal tasks delineated, the ultimate objective comprises the sequential completion of four salient subtasks: (1) opening the microwave, (2) relocating the kettle, (3) toggling the light switch, and (4) initiating the sliding action of the cabinet door. In conjunction, three comprehensive datasets have been provisioned. The '-complete' dataset encompasses demonstrations where all four target subtasks are executed in a sequential manner. The '-partial' dataset features various tasks, but it distinctively includes sub-trajectories wherein the aforementioned four target subtasks are sequentially achieved. The '-mixed' dataset captures an assortment of subtask executions; however, it is noteworthy that the four target subtasks are not completed in an ordered sequence within this dataset. We utilize these datasets in our experiments.

## A.2  HYPER-PARAMETERS AND IMPLEMENTATIONS

Table 3: ADT Actor (Transformer) Hyper-parameters

|  | Hyper-parameter | Value |
|---|---|---|
| Architecture | Hidden layers | 3 |
| | Hidden dim | 128 |
| | Heads num | 1 |
| | Clip grad | 0.25 |
| | Embedding dim | 128 |
| | Embedding dropout | 0.1 |
| | Attention dropout | 0.1 |
| | Residual dropout | 0.1 |
| | Activation function | GeLU |
| | Sequence length | 20 (V-ADT), 10 (G-ADT) |
| | G-ADT Way Step | 20 (kitchen-partial, kitchen-mixed), 30 (Others) |
| Learning | Optimizer | AdamW |
| | Learning rate | 1e-4 |
| | Mini-batch size | 256 |
| | Discount factor | 0.99 |
| | Target update rate | 0.005 |
| | Value prompt scale | 0.001 (Mujoco) 1.0 (Others) |
| | Warmup steps | 10000 |
| | Weight decay | 0.0001 |
| | Gradient Steps | 100k (G-ADT, AntMaze), 1000k (Others) |

We provide the lower-level actor's hyper-parameters used in our experiments in Table 3. Most hyper-parameters are set following the default configurations in DT. For the inverse temperature used in calculating the AWR loss of the lower-level actor in V-ADT, we set it to 1.0, 3.0, 6.0, 6.0, 6.0, 15.0 for antmaze-'umaze', 'umaze-diverse', 'medium-diverse', 'medium-play', 'large-diverse', 'large-play'

dataset, respectively; for other datasets, it is set 3.0. As for G-ADT, the inverse temperature is set to 1.0 for all the datasets. For the critic used in V-ADT and G-ADT, we follow the default architecture and learning settings in IQL (Kostrikov et al., 2022) and HIQL (Park et al., 2023), respectively.

The implementations of ADT is based on CORL repository (Tarasov et al., 2022). A key different between the implementation of ADT and DT is that we follow the way in (Badrinath et al., 2023) that we concatenate the (scaled) prompt and state, then the concatenated information and the action are treated as two tokens per timestep. In practice, we pretrain the critic for ADT, then the critic is used to train the ADT actor. For each time of evaluation, we run the algorithms for 10 episodes for MuJoCo datasets, 50 episodes for Kitchen datasets, and 100 episodes for AntMaze datasets.

## B   IQL AND HIQL

Implicit Q-learning (IQL) (Kostrikov et al., 2022) offers a nuanced approach to managing out-of-sample action queries. This is achieved by transforming the traditional max operator in the Bellman optimality equation to an expectile regression framework. More formally, IQL constructs an action-value function $Q(s, a)$ and a corresponding state-value function $V(s)$. These are governed by the loss functions:

$$\mathcal{L}_V = \mathbb{E}_{(s,a)\sim\mathcal{D}} \left[ L_2^\tau \left( \bar{Q}(s, a) - V(s) \right) \right], \tag{7}$$

$$\mathcal{L}_Q = \mathbb{E}_{(s,a,s')\sim\mathcal{D}} \left[ \left( r(s, a) + \gamma V(s') - Q(s, a) \right)^2 \right], \tag{8}$$

Here, $\bar{Q}$ symbolizes the target Q network, and $L_2^\tau$ is defined as the expectile loss with a parameter constraint $\tau \in [0.5, 1)$ and is mathematically represented as $L_2^\tau(x) = |\tau - \mathbb{1}(x < 0)|x^2$.

Building on this foundation, Hierarchical Implicit Q-Learning (Park et al., 2023) introduces an action-free variant of IQL that facilitates the learning of an offline goal-conditioned value function $V(s, g)$:

$$\mathcal{L}_V = \mathbb{E}_{(s,s')\sim\mathcal{D}_\mathcal{S}, g\sim p(g|\tau)} \left[ L_2^\tau \left( r(s, g) + \gamma \bar{V}(s', g) - V(s, g) \right) \right] \tag{9}$$

where $\bar{V}$ denotes the target Q network. Then a high-level policy $\pi_h^h(s_{t+k} \mid s_t, g)$, which produces optimal $k$-step subgoals $s_{t+k}$ is trained via:

$$J_{\pi^h} = \mathbb{E}_{(s_t, s_{t+k}, g)} \left[ \exp\left( \beta \cdot \tilde{A}^h(s_t, s_{t+k}, g) \right) \log \pi_h^h(s_{t+k} \mid s_t, g) \right], \tag{10}$$

where $\beta$ represents the inverse temperature hyper-parameter, and the value $\tilde{A}^h(s_t, s_{t+k}, g)$ is approximated using $V(s_{t+k}, g) - V(s_t, g)$. For a comprehensive exploration of the methodology, readers are encouraged to consult the original paper.

## C   PERFORMANCE GAP WITH ANTICIPATED UPPER BOUND

While ADT has demonstrated superior performance relative to baseline models, the overall performance lags behind MLP-based offline RL algorithms, e.g, IQL. Thus we hope to probe ADT's potential in approaching a learning upper bound of transformer-based offline RL algorithms. To this end, we initiate our analysis by pre-training an IQL approach. In addition to substituting the target return with IQL's value-function, the action adopted at every state in the dataset is also relabeled using IQL's actor. Drawing upon the proficiency of IQL in trajectory stitching, the re-labeled data can seamlessly incorporate more near-optimal sub-trajectories. In training with this refined data, ADT benefits in two primary ways: first, it can learn to stitch sub-trajectories under the guidance of Q-values; secondly, it can directly imitate the refined sub-trajectories. Consequently, we posit that this methodology can act as an oracle to approach the empirical learning upper bound of ADT. As illustrated in Figure 6, V-ADT shows a resemblance to that of V-ADT Oracle's performance on certain datasets. Nevertheless, deviations are evident in some, indicating a further refining of the training procedures to achieve optimal outcomes should be investigated as future work.

## D   COMPLETE EXPERIMENTAL RESULTS

Here we provide the learning curves of our methods on all selected datasets.

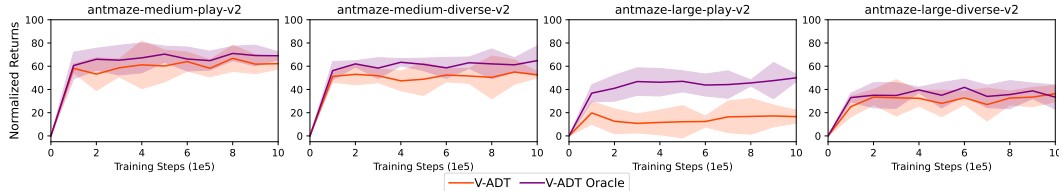

Figure 6: Learning curves of V-ADT and V-ADT Oracle. The performance gap between V-ADT and V-ADT Oracle is limited on some datasets while evident on others, requiring further refinement.

Figure 7: Learning curves of V-ADT.

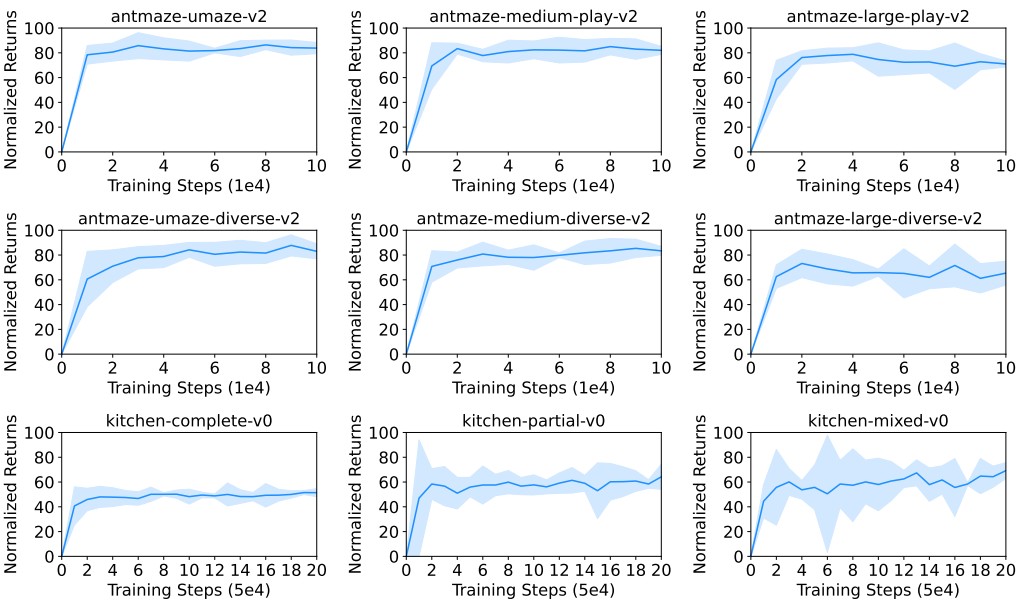

Figure 8: Learning curves of G-ADT.

# E    VISUALIZATION OF DECISION-MAKING PROCESS OF G-ADT

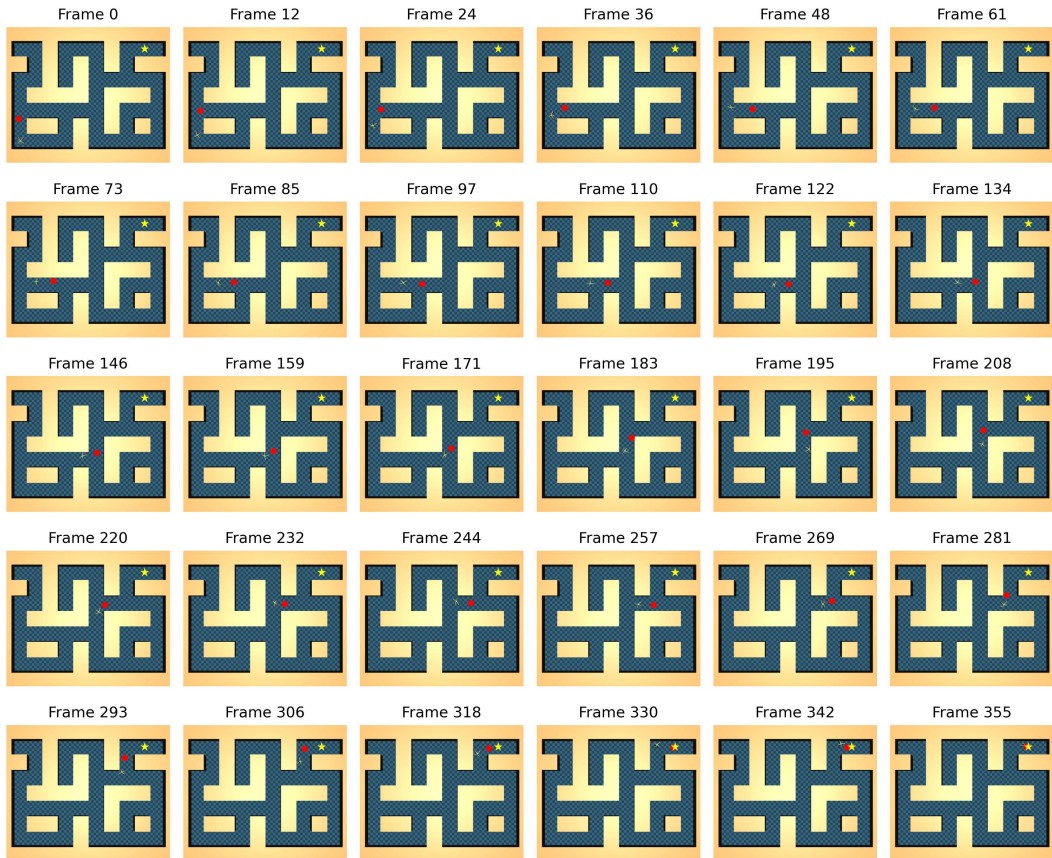

Figure 9: Example of decision-making process of G-ADT in antmaze-large-play-v2 environments. We present some snapshots within an episode. The red circle represents the sub-goal given by the prompt policy. The pentagram indicates the target position to arrive.

