# OpenReview forum: "Rethinking Decision Transformer via Hierarchical Reinforcement Learning"
_ICLR.cc/2024/Conference — Submitted to ICLR 2024_

### Official Review · Reviewer_UBLF · 2023-10-25

**Soundness:** 3 good
**Presentation:** 3 good
**Contribution:** 2 fair
**Rating:** 5
**Confidence:** 4

**Summary:**

This study introduces the Autotuned Decision Transformer (ADT), a novel approach that employs a hierarchical structure, substituting the traditional returns-to-go (RTG) with prompts derived from a high-level policy. The paper presents two specific variants of this innovative prompting mechanism: V-ADT, which utilizes prompts designed to optimize learned value functions; G-ADT, where the prompts provide subgoals, strategically directing the policy toward the ultimate objective. Proposed methods demonstrates superior performance compared to conventional DT-based techniques and hierarchical methods on the standard D4RL benchmarks.

**Strengths:**

**Strength 1: Effective Approach to a Critical Issue**

This paper addresses a crucial issue in the realm of Decision Transformers (DT), particularly the challenge associated with handling the returns-to-go (RTG) and integrating value functions during the DT training phase. There is widespread agreement on the necessity of this challenge within the field. The proposed solution, which involves learning a prompt to feed into the policy of the DT, is not only a plausible approach but also one that has been empirically substantiated, showing enhanced effectiveness over existing DT-based methodologies. This validation underscores the method's potential impact and applicability within the discipline.

**Strength 2: Noteworthy Innovation in Synthesis**

While the individual components utilized in the proposed method might not be pioneering in isolation—such as the employment of in-sample optimal values (akin to value-based approaches like IQL), the adoption of hierarchical structures for subgoals (seen in strategies like HIQL), or the application of weighted regression techniques (as in AWR)—the paper's true innovation lies in the synthesis of these elements within the framework of Decision Transformers.

The authors have skillfully amalgamated these various concepts, producing a methodology that, in its entirety, presents a significant departure from conventional approaches. This fusion of ideas, culminating in a cohesive and well-articulated study, embodies substantial novelty.

**Strength 3: Insightful Ablations**

Given the many components of the proposed method, it's important to empirically validate the source of the gain the proposed method may offer. In this sense, the authors made a great effort on additional experiments from page 7 to 9.

**Weaknesses:**

**Weakness 1: Limited Empirical Superiority**

The primary data presented in Table 1 indicate that while V-ADT demonstrates a notable advancement over other DT-based methods, it fails to consistently surpass or significantly differentiate itself from value-based strategies. This observation is critical, particularly since V-ADT incorporates a value function trained via IQL during the formation of its high-level policy, suggesting that its performance should be directly contrasted with these value-based counterparts. The authors' choice to emphasize the superiority of V-ADT within the DT category, marked in bold, might inadvertently misguide readers regarding the method's comparative effectiveness. It remains unclear why V-ADT, despite leveraging the advantages of pre-trained IQL values, does not achieve a substantial performance lead over IQL itself. A more in-depth discussion on this aspect would enhance the paper's credibility and help readers understand the practical implications and potential limitations of integrating value-based components within a DT framework.

**Weakness 2: Insufficiency of Information for Replicability**

The absence of shared code alongside the paper significantly hampers the research's transparency and the reproducibility of its findings. The explanation provided in Appendix A, particularly the vague descriptions in the concluding section, lacks the detailed guidance necessary for readers to independently replicate and verify the proposed method's effectiveness. Considering the empirical foundation upon which the method stands, it is imperative for the authors to release the corresponding code, ensuring that peers can thoroughly evaluate, validate, or even potentially improve upon the methodology.

**Questions:**

Question 1) The comprehensive ablation studies provided are certainly insightful. Could the authors elaborate on their specific choice of using antmaze for these ablations (not MuJoCo), considering its characteristic sparse rewards and diverse objectives?

---

> ### Author Response · Authors · 2023-11-14
> **Response to Reviewer UBLF**
>
> ### Q: Limited Empirical Superiority
>
> A:  Our ablation studies reveal some interesting findings that might be able to explain why the overall performance of ADT falls short of IQL and HIQL. **Our main conjecture is that Transformer as a function approximator is hard to optimize for standard RL algorithms.** To verify this, in Section C in the appendix of the revised paper, we first implement an *oracle algorithm*, which distill the IQL policy using a transformer with supervised learning objective (V-ADT Oracle in Figure 6). The oracle algorithm matches the performance of IQL, suggesting that the transformer architecture is not the bottleneck. We then implement another baseline, named IQL-Trans , by replacing the MLP policy with a transformer policy for IQL. As shown in Figure 3, the performance of IQL-Trans (V-ADT w/o prompt in Figure 3) cannot match the original IQL, further supporting our conjecture. The advantage of ADT over IQL-Trans is mainly contributed to the joint optimization of hierarchical policies (the high-level policy optimizes the prompt and the the low-level policy is optimized based on the prompt), since this is the key difference between these two algorithms. To help readers clearly get this information, we provide the results of V-ADT, V-ADT Oracle, IQL-Trans and IQL in the table below:
>
> |                           |     V-ADT      |   IQL-Trans    |  V-ADT Oracle  |       IQL       |
> | :------------------------ | :------------: | :------------: | :------------: | :-------------: |
> | antmaze-medium-play-v2    | 62.2 $\pm$ 2.5 | 50.6 $\pm$ 6.6 | 69.0 $\pm$ 1.8 | 71.2 $\pm$ 7.3  |
> | antmaze-medium-diverse-v2 | 52.6 $\pm$ 1.4 | 38.6 $\pm$ 5.4 | 64.8 $\pm$ 6.5 | 70.0 $\pm$ 10.9 |
> | antmaze-large-play-v2     | 16.6 $\pm$ 2.9 | 19.4 $\pm$ 3.6 | 50.0 $\pm$ 1.7 | 39.6 $\pm$ 5.8  |
> | antmaze-large-diverse-v2  | 36.4 $\pm$ 3.6 | 5.0 $\pm$ 5.2  | 33.4 $\pm$ 5.3 | 47.5 $\pm$ 9.5  |
>
> Finally, we note that there is still a performance gap between ADT and the oracle algorithm. This motivates the investigation of other techniques to improve  transformer-based decision models, which we leave as our future work.
>
>
>
> ### Q: Insufficiency of Information for Replicability
>
> A: We promise that we will release our codes soon.
>
>
>
> ### Q: why using antmaze for these ablations (not MuJoCo), considering its characteristic sparse rewards and diverse objectives?
>
> A: One of the most important ADT's contribution is giving transformer-based models (DT) the stitching ability, referring to the capability to integrate suboptimal trajectories from the data. The Antmaze datasets consist of suboptimal trajectories, requiring the agent use portions of existing trajectories in order to solve a task, is widely considered to be the benchmark for evaluating the stitching ability [1]. Therefore, we choose to ablate the efficacy of each component of ADT in improving the stitching ability using Antmaze datasets.
>
> ### Reference
>
> [1] Fu, Justin, Aviral Kumar, Ofir Nachum, George Tucker, and Sergey Levine. "D4rl: Datasets for deep data-driven reinforcement learning." arXiv preprint arXiv:2004.07219 (2020).

---

> > ### Comment · Reviewer_UBLF · 2023-11-21
> >
> > I appreciate the authors for the response, which has helped clarify several aspects of the work.
> >
> > I now have a better understanding of the rationale behind selecting the antmaze environment for ablations and its relevance in evaluating the stitching ability. This aspect of the research is well-justified and adds value to the study.
> >
> > Regarding the difficulty in optimizing the Transformer with standard offline RL algorithms, I agree with the author's response. However, it is noteworthy that even with the implementation of the V-ADT Oracle, its performance lags behind IQL in the Antmaze environment. This observation leads to a lack of compelling reasons for readers to prefer V-ADT over IQL in this specific context.
> >
> > Consequently, while I acknowledge the potential and innovativeness of the hierarchical transformer-based approach, these current limitations lead me to maintain my initial rating. Nevertheless, I see significant promise in the methodology. I encourage the authors to improve the approach, particularly in bridging the performance gap observed.

---

> > > ### Author Response · Authors · 2023-11-21
> > >
> > > Thank you for your response. We appreciate the recognition of our work's potential and innovativeness.
> > >
> > > To clarify some potential misunderstandings, we would like to point out that V-ADT Oracle should be understood as a "Transformer for RL Oracle". We pick IQL as an oracle algorithm, and distill its policy using  a transformer. What the gap between this oracle and IQL really characterizes is the limitations of using a transformer policy in "AntMaze" and "Kitchen", not the limitations of  V-ADT. We also note that this conclusion is not universal. For MuJoCo Control problems as shown in the table below, the oracle outperforms IQL, showing the advantages of using a transformer policy in these domains. V-ADT also outperforms IQL and almost matches the oracle. More importantly, V-ADT significantly outshines other methods employing transformer policies (like DT, QLDT), proving that its potential stems from its design rather than solely its transformer policy basis.
> > >
> > > |                           | V-ADT | V-ADT Oracle | IQL   |
> > > | ------------------------- | ----- | ------------ | ----- |
> > > | walker2d-medium           | 80.9  | 82.5         | 78.3  |
> > > | walker2d-medium-replay    | 86.3  | 78.4         | 73.8  |
> > > | walker2d-medium-expert    | 112.1 | 107.9        | 109.6 |
> > > | hopper-medium             | 60.6  | 70.1         | 66.2  |
> > > | hopper-medium-replay      | 83.5  | 90.9         | 94.7  |
> > > | hopper-medium-expert      | 101.6 | 97.1         | 91.5  |
> > > | halfcheetah-medium        | 48.7  | 48.4         | 47.4  |
> > > | halfcheetah-medium-replay | 42.8  | 43.9         | 44.2  |
> > > | halfcheetah-medium-expert | 91.7  | 92.2         | 86.7  |
> > > | Total                     | 708.3 | 711.4        | 692.1 |
> > >
> > >
> > >
> > > Our observations suggest that for  "AntMaze" and "Kitchen", we might need some specific designs for the transformer architecture.  Existing research also illustrates the variable effectiveness of Transformers compared to MLPs in offline RL benchmarks. For instance, as Table 1 in RvS[1] indicates, DT falls short of RvS-R in Antmaze datasets, yet surpasses it in MuJoCo within the reward-conditioned behavior cloning framework. This variability aligns with our findings that the superiority of using either Transformers or MLPs in single-task is domain-dependent.   We believe this presents an intriguing research avenue for future exploration, yet it is orthogonal to our primary contributions.
> > >
> > >  If you find our reply addresses your concerns about the performance of V-ADT and algorithm choice in different domains, we would be grateful if you could reconsider and potentially increase your score.
> > >
> > >
> > >
> > > ### Reference
> > >
> > > [1] Emmons, Scott, Benjamin Eysenbach, Ilya Kostrikov, and Sergey Levine. "RvS: What is Essential for Offline RL via Supervised Learning?." In *International Conference on Learning Representations*. 2021.

---

> > > > ### Author Response · Authors · 2023-11-22
> > > >
> > > > Dear reviewer, we have provided detailed responses, but have not yet hear back from you. We will appreciate it deeply if you could reply our rebuttal. We are sincerely looking forward to further discussions to address the reviewers concerns to our best. Thanks!

---

### Official Review · Reviewer_xoB5 · 2023-10-30

**Soundness:** 2 fair
**Presentation:** 2 fair
**Contribution:** 2 fair
**Rating:** 5
**Confidence:** 3

**Summary:**

This paper investigates transformer-based decision models through a hierarchical decision-making framework, and proposes two new transformer-based decision models for offline RL. Specifically, the high-level policy suggests a prompt, following which a low-level policy acts based on this suggestion. Empirical studies show some improvements over DT on several control and navigation benchmarks.

**Strengths:**

The study of integrating decision transformers into a hierarchical setting is interesting to the HRL community, for understanding the benefits as well as limitations of the DT based approach. The two proposed models seem to be technically sound. The empirical studies seem to be comprehensive.

**Weaknesses:**

1. Integrating decision transformers into a hierarchical setting has already been studied in an earlier paper [*], especially corresponding to the goal-conditioned version. This paper wasn’t cited or discussed.
2. The paper doesn’t dive into the analysis of the reason why the proposed approach outperforms or underperforms the baseline methods. For instance, when comparing with HRL baselines, it was simply mentioning that “Given that V-ADT and G-ADT is trained following the IQL and HIQL paradigm, respectively, the achieved performance nearing or inferior to that of IQL and HIQL is anticipated” - but why?  Is it due to the generated subgoals or non-stationarity issues? The reasons behind the observations are more valuable to the community.
3. In V-ADT, does the high level provide prompt at every time step? If it does, it is arguably a hierarchical setting since there is no clear decomposition of a global goal or task. What is the reward function of the high level?
4. It’s not clear to me how the high level generates returns which are not covered in the offline dataset either? i.e., how could you avoid the issues you stated in Sec. 2
Reference:
[*] Correia, A. and Alexandre, L.A., 2022. Hierarchical decision transformer. arXiv preprint arXiv:2209.10447.

**Questions:**

Please address my questions in the Weakness section

---

> ### Author Response · Authors · 2023-11-14
> **Response to Reviewer xoB5: Part 1**
>
> ### Q: Integrating decision transformers into a hierarchical setting has already been studied in an earlier paper
>
> A: Thanks for pointing this out. We have cited and discussed this paper and other DT Enhancements in Discussions section in our revised paper. ADT differs from HDT mainly from two aspects: (1) The high-level policy of ADT is more general that could output different kinds of prompts, including sub-goals, target values, latent skills and options. (2) For the low-level policy, we apply advantage weighted regression instead of conditioned prediction as in DT to enable stitching ability.
>
> ### Q: why the proposed approach outperforms or underperforms the baseline methods
>
> A: Thanks for your suggestion! Our ablation studies are designed to explain the reasons behind the empirical results. But we totally agree that analysis of these results should be discussed more carefully in the paper.
>
> Our ablation studies reveal some interesting findings that might be able to explain why the overall performance of ADT falls short of IQL and HIQL. **Our main conjecture is that Transformer as a function approximator is hard to optimize for standard RL algorithms.** To verify this, in Section C in the appendix of the revised paper, we first implement an *oracle algorithm*, which distill the IQL policy using a transformer with supervised learning objective (V-ADT Oracle in Figure 6). The oracle algorithm matches the performance of IQL, suggesting that the transformer architecture is not the bottleneck. We then implement another baseline, named IQL-Trans , by replacing the MLP policy with a transformer policy for IQL. As shown in Figure 3, the performance of IQL-Trans (V-ADT w/o prompt in Figure 3) cannot match the original IQL, further supporting our conjecture. The advantage of ADT over IQL-Trans is mainly contributed to the joint optimization of hierarchical policies (the high-level policy optimizes the prompt and the the low-level policy is optimized based on the prompt), since this is the key difference between these two algorithms. To help readers clearly get this information, we provide the results of V-ADT, V-ADT Oracle, IQL-Trans and IQL in the table below:
>
> |                           |     V-ADT      |   IQL-Trans    |  V-ADT Oracle  |       IQL       |
> | :------------------------ | :------------: | :------------: | :------------: | :-------------: |
> | antmaze-medium-play-v2    | 62.2 $\pm$ 2.5 | 50.6 $\pm$ 6.6 | 69.0 $\pm$ 1.8 | 71.2 $\pm$ 7.3  |
> | antmaze-medium-diverse-v2 | 52.6 $\pm$ 1.4 | 38.6 $\pm$ 5.4 | 64.8 $\pm$ 6.5 | 70.0 $\pm$ 10.9 |
> | antmaze-large-play-v2     | 16.6 $\pm$ 2.9 | 19.4 $\pm$ 3.6 | 50.0 $\pm$ 1.7 | 39.6 $\pm$ 5.8  |
> | antmaze-large-diverse-v2  | 36.4 $\pm$ 3.6 | 5.0 $\pm$ 5.2  | 33.4 $\pm$ 5.3 | 47.5 $\pm$ 9.5  |
>
> Finally, we note that there is still a performance gap between ADT and the oracle algorithm. This motivates the investigation of other techniques to improve  transformer-based decision models, which we leave as our future work.

---

> > ### Comment · Reviewer_xoB5 · 2023-11-22
> >
> > I appreciate the authors' efforts in addressing the concerns raised in my initial review. However, I still have reservations regarding the distinction between this work and the prior work of the Hierarchical Decision Transformer.
> >
> > The authors' response suggests that the high-level policy of ADT is more general, capable of outputting various types of prompts. This statement implies that the prior work might be limited to certain types of subgoals. For a clearer understanding of the novelty and contributions of this paper, it would be beneficial to have a more detailed explanation of how this generalization differs from or improves upon the capabilities of the Hierarchical Decision Transformer.
> >
> > Additionally, I noticed that the revised manuscript does not include direct experimental comparisons with the Hierarchical Decision Transformer. Such comparisons are crucial for substantiating the claimed advancements and understanding the relative strengths and limitations of both approaches.
> >
> > I believe that further clarifications and possibly additional experimental comparisons are needed to fully address these concerns. These additions would significantly strengthen the paper and provide the necessary context for its positioning within the existing body of work.

---

> > > ### Author Response · Authors · 2023-11-22
> > >
> > > Thanks for the further discussion!
> > >
> > > ### Q1: More detailed explanation of how this generalization differs from or improves upon the capabilities of the Hierarchical Decision Transformer.
> > >
> > > A: The prompt used in HDT is the sub-goal (i.e., a future state in the datasets) for the low-level policy to reach. While the prompt used in ADT can be seen as a latent action generated by the high-level policy, serving as guidance for the low-level policy to inform its decision-making process.  This latent action could include several types of prompts including return-to-go, value, sub-goal, skills, options and so on.  In this case, HDT could be recovered as a special case of ADT's framework in regards of the high-level prompt generation.
> > >
> > > In addition to the generalized prompt generation, the novelty and contributions of ADT also lies in the investigation of how to jointly optimize the high-level and low-level policies to enable the stitching capability of transformer-based policy. The introducing of training transformer-based policy using RL is also a vital component of ADT for achieving superior performance.
> > >
> > > In conclusion, for the high-level, HDT could be regarded as a special case of ADT; for the low-level, the RL training used in ADT bring more benefits than behavioral cloning used in HDT in improving the stitching ability.
> > >
> > >
> > >
> > >
> > >
> > > ### Q2: experimental comparisons with the Hierarchical Decision Transformer.
> > >
> > > A: As you suggested, we provide comparison between HDT and ADT in the table below. As HDT reports the best score, to ensure fair comparison, we report best normalized scores  for both HDT and V-ADT.  The results of HDT are directly taken and transferred to the normalized score using the function provided in d4rl package. As shown in the table, except for hopper-medium, ADT outperforms HDT on all datasets.
> > >
> > > |                    | HDT  | V-ADT     |
> > > | ------------------ | ---- | --------- |
> > > | halfcheetah-medium | 44.2 | **49.9**  |
> > > | hopper-medium      | 95.0 | 81.3      |
> > > | walker2d-medium    | 84.5 | **89.5**  |
> > > | kitchen-complete   | 65.0 | **66.0**  |
> > > | maze2d-medium      | 66.2 | **120.2** |
> > >
> > >
> > >
> > > We  hope our reply addresses your confusion of the difference and the connection between ADT and HDT. We would be grateful if you could reconsider and potentially increase your score. Thanks!

---

> ### Author Response · Authors · 2023-11-14
> **Response to Reviewer xoB5: Part 2**
>
> ### Q: does the high level provide prompt at every time step?
>
> A: Yes, we describe this in Figure 1.
>
>
>
> ### Q: V-ADT is arguably a hierarchical setting since there is no clear decomposition of a global goal or task
>
> A: The term "hierarchical" is mainly referred to the policy decomposition $\pi(a|s) = \int_{\rho\in \mathcal{P}} \pi(\rho | s) \pi(a|s,\rho) d \rho$  (Eq 3). The prompt is a general concept, includes but not limits to sub-goals used in HRL. Other choices includes returns-to-go (values), option, and skills. For example, return-conditioned prediction [1] can also be included as a special case of hierarchical policy framework.
>
> Although V-ADT does not decompose a global goal into explicit sub-goals, it operates hierarchically by having the high-level policy generate context-specific prompts (values in this case) which guide the decision-making of the low-level policy. This separation of prompt generation (high-level) and action determination (low-level) based on these prompts aligns with the fundamental concept of hierarchy in decision-making processes.
>
>
>
> ### Q: What is the reward function of the high level?
>
> A: There is no reward function of the high-level in V-ADT. The low-level policy of V-ADT makes return-conditioned predictions as in [1]. The key innovation of V-ADT is the value prompt given by the high-level policy is learned from data.
>
>
>
> ### Q: how could you avoid the issues you stated in Sec. 2?
>
>  V-ADT can solve the motivating example. The value prompt used by V-ADT represents the maximum achievable value given the offline data. This value prompt then informs the V-ADT low-level policy to produce an action towards achieving the value.
>
> Given this in mind, let's consider the motivating example.  The dataset \( D \) contains three trajectories $a → b → c$ leading to a return of 0, $d → b → e$ leading to a return of 10, and another trajectory starting from *$a$* and leading to a return of 1. As for this particular example, the value prompt for $a$ is 10. To achieve this value, at state $a$, V-ADT takes a step towards $b$, as the other path would lead a worse trajectory with return 1.
>
>
>
>
>
> ### Reference
>
> [1] Emmons, Scott, Benjamin Eysenbach, Ilya Kostrikov, and Sergey Levine. "RvS: What is Essential for Offline RL via Supervised Learning?." In *International Conference on Learning Representations*. 2021.

---

> > ### Author Response · Authors · 2023-11-22
> >
> > Dear reviewer, we have provided detailed responses, but have not yet hear back from you. We will appreciate it deeply if you could reply our rebuttal. We are sincerely looking forward to further discussions to address the reviewers concerns to our best. Thanks!

---

### Official Review · Reviewer_XUDW · 2023-10-31

**Soundness:** 3 good
**Presentation:** 3 good
**Contribution:** 2 fair
**Rating:** 6
**Confidence:** 4

**Summary:**

This paper presents an in-depth examination of a hierarchical reinforcement learning (HRL) approach applied to decision transformers (DT). The study begins by shedding light on various challenges within the realm of DT, primarily focusing on issues like inaccurate return-to-go (RTG) estimates resulting from initial estimations and the inability perform stitching. The authors propose a solution by introducing a simple network, conditioned solely on the current state, which generates a prompt for the high-level policy. Furthermore, they refine the low-level policy by conditioning it on both the state, historical data (excluding returns), and the generated prompt to enhance in context learning.  Advantage regression is used to train the low-level policy since the value tokens are not conditioned on the actions. The paper explores two distinct prompt styles: one based on value, learned using in-sample optimal value, to address the RTG estimation issues, and another that outputs a goal state prompt, learned using HIQL.

The experimental component encompasses a variety of D4RL environments, with particular emphasis on hierarchical envs. The results show improvements mainly in the hierarchical environments. It also shows issues related to variance when tuning target returns, showcasing consistent performance using the proposed approaches. Additionally, the paper provides insights through various ablations, which involve removing RL losses and prompts, among other factors.

**Strengths:**

- This work gives a well-thought-out approach compared to most existing research involving HRL in the context of decision transformers. It addresses specific challenges, such as the problems associated with RTG estimates, in a systematic manner, such as by introducing value tokens.
- The introduction of value-based prompts represents a novel contribution, and the adjustment of the low-level policy loss to accommodate this is a good improvement.
- The paper includes a substantial number of offline baselines and conducts numerous experiments in hierarchical environments, as well as those involving complex stitching operations.
- The ablation studies effectively illustrate the significance of each component within the proposed approach.
- The paper is easy to read and very well written.

**Weaknesses:**

I have two main concerns about this work:

- The experimental results presented in the paper appear to lack significance. For instance, in Table 2, the performance of G-ADT is nearly indistinguishable from that of the waypoint transformer. In Table 1, V-ADT demonstrates significant improvements over DT only in the of antmaze environments, with results still falling short of IQL.
- While the application of various RL losses to DT within a hierarchical framework has elements of novelty, the significance of these contributions may be undermined by the underwhelming experimental outcomes.

In light of the concerns regarding the lack of experimental significance, it is recommended to consider a weak rejection. The work remains principled and thought-provoking; however, additional experimental evaluations are essential to further substantiate its claims and contributions.

**Questions:**

Can more HRL environment experiments be shown?

Can the authors put the significance of the work into context better (especially experimentally)?

---

> ### Author Response · Authors · 2023-11-14
> **Response to Reviewer XUDW**
>
> ### Q: The experimental results presented in the paper appear to lack significance
>
> A: We respectfully disagree with this. V-ADT outperforms DT by **20.5%** on MuJoCo datasets, **227.5%** on Antmaze datasets, and **42.5%** on Kitchen datasets.  G-ADT  outperforms WT by **14.2%** on Antmaze datasets.
>
> The overall performance of ADT falls short of IQL could be attributed to the transformer architecture of the policy. In fact, it is still an open problem if Transformer is more efficient than standard architectures in single task RL. Previous studies show that MLP is competitive with and sometimes more effective than Transformer in offline RL benchmarks (see Table 1 of RvS[1] for example, where RvS-R outperforms DT on Antmaze datasets in the reward-conditioned behavior cloning setting). One conjecture is that high-capacity modles only make sense when we train on large and diverse datasets [2]. We would like to explore this conjecture in future works by applying ADT for multi-modal and multi-task domains.
>
>
>
> ### Q: Can more HRL environment experiments be shown?
>
> A: Our method is not designed specifically for HRL. Instead, we use hierachical framework to rethink transformer-based decision algorithms, and derive joint policy optimization algorithms from it. Our method is essentially an offline RL method and we follow standard benchmarks to prove the effectiveness of ADT.
>
> ### Reference
>
> [1] Emmons, Scott, Benjamin Eysenbach, Ilya Kostrikov, and Sergey Levine. "RvS: What is Essential for Offline RL via Supervised Learning?." In *International Conference on Learning Representations*. 2021.
>
> [2] Chebotar, Y., Vuong, Q., Hausman, K., Xia, F., Lu, Y., Irpan, A., ... & Levine, S. (2023, August). Q-Transformer: Scalable Offline Reinforcement Learning via Autoregressive Q-Functions. In *7th Annual Conference on Robot Learning*.

---

> ### Comment · Reviewer_XUDW · 2023-11-21
> **Response**
>
> I thank the authors for the clarification.  I also believe the authors have given good justification for the IQL performance gap in my response and to other reviewer responses.  I have chosen to adjust my score to marginal acceptance.

---

### Official Review · Reviewer_RLH1 · 2023-11-02

**Soundness:** 2 fair
**Presentation:** 2 fair
**Contribution:** 2 fair
**Rating:** 5
**Confidence:** 5

**Summary:**

This paper proposes to integrate a hierarchical architecture with the decision transformer architecture to automatically tune the “prompts”. The prompt in this paper mainly corresponds to the reward-to-go in the original DT paper, and is also extended to the notion of “goal”. The paper proposed an automated decision transformer (ADT), and its two versions, including V-ADT and G-ADT. Experimental results in continuous control tasks show that ADT shows better performances than baselines.

**Strengths:**

- The paper focuses on a solid problem which is manually selecting a pre-defined expected reward is often difficult and can result in suboptimal outcomes.

 - The ablation study is interesting.

**Weaknesses:**

- The paper is missing key related works, such as prompt decision transformer studying the prompting mechanism for continuous control tasks in multi-task setting and hierarchical decision transformer, which has a similar motivation of using a hierarchical architecture.

    - Prompting decision transformer for few-shot generalization: https://arxiv.org/pdf/2206.13499.pdf

    - Hierarchical decision transformer: https://arxiv.org/pdf/2209.10447.pdf

 - Although the problem is quite solid, and the paper is trying to propose a general hierarchical framework to tackle the problem, the paper is still focusing on single-task learning, which is a relatively narrow scope.

 - The methodology is hard to follow, and the design choice is not fully discussed.

 - The motivating example in Section 2.2.1 does not quite connect to the proposed methodology and leads to confusion. The example provided in section 2.2.1 is mainly about the data coverage issue. The proposed method seems to still learning Q, V, or goal from the offline dataset. How the proposed method can help solve the problem is still unclear.

**Questions:**

- For the hierarchical policy, why condition on the current state is sufficient?

 - Does the method’s performance heavily depend on the learned Q and V function from IQL?

 - How the proposed method can help solve the example problem in section 2.2.1 (no trajectory a->b->c)?

 - How is the proposed method different from the hierarchical decision transformer?

---

> ### Author Response · Authors · 2023-11-14
> **Response to Reviewer RLH1**
>
> ### Q: The paper is missing key related works
>
> A: Thanks for pointing out this. We have cited and discussed these two papers and other DT Enhancements in Discussions section in our revised paper
>
>
>
> ### Q: The paper is still focusing on single-task learning.
>
> A: Actually, some of  the tasks in our experiments include several sub-tasks. For example, the kitchen datasets are collected based on the 9 degrees of freedom [Franka robot](https://www.franka.de/). The Franka robot is placed in a kitchen environment containing several common household items: a microwave, a kettle, an overhead light, cabinets, and an oven. The environment is a `multi-goal reaching` problem in which the robot has to interact with the previously mentioned items in order to reach a desired goal configuration. Detailed descriptions can be found at https://robotics.farama.org/envs/franka_kitchen/franka_kitchen/ and https://github.com/Farama-Foundation/d4rl/wiki/Tasks#frankakitchen. We leave multi-task learning on more benchmarks as our future work.
>
>
>
> ### Q: The methodology is hard to follow, and the design choice is not fully discussed.
>
> A: To help understand our methodology better, we provide more detailed descriptions in Section 3 of the revision. As for the design choices, we ablate the effectiveness of the most important components of ADT including the efficacy of the prompt, the efficacy of the RL loss and the efficacy of tokenization in Section 5.4. If you could please point out the specific design choices that need to be discussed, we would happy to respond to them.
>
>
>
> ### Q: The motivating example in Section 2.2.1 does not quite connect to the proposed methodology and leads to confusion.  How the proposed method can help solve the example problem in section 2.2.1 (no trajectory a->b->c)?
>
> A: V-ADT can solve the motivating example. The value prompt used by V-ADT represents the maximum achievable value given the offline data. This value prompt then informs the V-ADT low-level policy to produce an action towards achieving the value.
>
> Given this in mind, let's consider the motivating example.  The dataset \( D \) contains three trajectories $a → b → c$ leading to a return of 0, $d → b → e$ leading to a return of 10, and another trajectory starting from *$a$* and leading to a return of 1. As for this particular example, the value prompt for $a$ is 10. To achieve this value, at state $a$, V-ADT takes a step towards $b$, as the other path would lead a worse trajectory with return 1.
>
>
>
> ### Q: For the hierarchical policy, why condition on the current state is sufficient?
>
> A: We think a memoryless policy is sufficient for the Markov environment. However, we certainly agree that applying an auto-regressive model for the high-level policy as did in HDT is also possible. We would like to explore this design in our future work. Thanks for your suggestions.
>
>
>
> ### Q: Does the method’s performance heavily depend on the learned Q and V function from IQL?
>
> A: Yes, V-ADT requires joint optimization of high and low level policies. The output of the high-level policy, which is produced by IQL, significantly affects the performance of V-ADT.
>
> We want to emphasize that this does not mean that V-ADT is  simply reinventing IQL by replacing the policy with a transformer. Our ablation shows that leveraging the prompts from a high-level policy significantly boost the performance of V-ADT on some datasets.
>
>
>
> ### Q: How is the proposed method different from the hierarchical decision transformer?
>
> A: ADT differs from HDT mainly from two aspects: (1) The high-level policy of ADT is more general that could output different kinds of prompts, including sub-goals, target values, latent skills and options. (2) For the low-level policy, we apply advantage weighted regression instead of conditioned prediction as in DT to enable stitching ability.

---

> > ### Author Response · Authors · 2023-11-22
> >
> > Dear reviewer, we have provided detailed responses, but have not yet hear back from you. We will appreciate it deeply if you could reply our rebuttal. We are sincerely looking forward to further discussions to address the reviewers concerns to our best. Thanks!

---

> > > ### Author Response · Authors · 2023-11-22
> > >
> > > To help better understand the superiority of ADT over HDT, we provide empirical comparison between HDT and ADT in the table below. As HDT reports the best score, to ensure fair comparison, we report best normalized scores  for both HDT and V-ADT.  The results of HDT are directly taken and transferred to the normalized score using the function provided in d4rl package. As shown in the table, except for hopper-medium, ADT outperforms HDT on all datasets.
> > >
> > > |                    | HDT  | V-ADT     |
> > > | ------------------ | ---- | --------- |
> > > | halfcheetah-medium | 44.2 | **49.9**  |
> > > | hopper-medium      | 95.0 | 81.3      |
> > > | walker2d-medium    | 84.5 | **89.5**  |
> > > | kitchen-complete   | 65.0 | **66.0**  |
> > > | maze2d-medium      | 66.2 | **120.2** |

---

### Meta-Review · Area_Chair_ySLa · 2024-01-10

**Metareview:**

This paper introduces a new framework that integrates HRL and Decision Transformers.
This is generally an interesting and timely topic.

The reviewers expressed concerns about the experimental evaluation, and in particular the lack of comparisons to Hierarchical Decision Transformer.
While the authors replied to this concern during the rebuttal (and generally improved the quality of the manuscript), this concern could not be fully addressed.

Overall I found the paper interesting and I want to encourage the authors to fully address the concerns of the reviewers. I am confident that, in the future, with a more thorough experimental evaluation, this paper will become a solid contribution to the literature.

**Justification For Why Not Higher Score:**

Although the paper is interesting, the reviewers expressed concerns about the experimental evaluation and connections withe existing literature. These concerns have not been fully addressed during the rebuttal.

**Justification For Why Not Lower Score:**

N/A

---

### Decision · Program_Chairs · 2024-01-16

Reject